# Pressure-tuned quantum criticality in the large-$D$ antiferromagnet DTN

Kirill Yu. Povarov [1] ✉, David E. Graf [2], Andreas Hauspurg[1,3],
Sergei Zherlitsyn [1], Joachim Wosnitza[1,3], Takahiro Sakurai [4], Hitoshi Ohta [5,6],
Shojiro Kimura[7], Hiroyuki Nojiri [7], V. Ovidiu Garlea [8], Andrey Zheludev[9],
Armando Paduan-Filho [10], Michael Nicklas [11] & Sergei A. Zvyagin [1] ✉

Strongly correlated spin systems can be driven to quantum critical points via various routes. In particular, gapped quantum antiferromagnets can undergo phase transitions into a magnetically ordered state with applied pressure or magnetic field, acting as tuning parameters. These transitions are characterized by $z = 1$ or $z = 2$ dynamical critical exponents, determined by the linear and quadratic low-energy dispersion of spin excitations, respectively. Employing high-frequency susceptibility and ultrasound techniques, we demonstrate that the tetragonal easy-plane quantum antiferromagnet $NiCl_2 \cdot 4SC(NH_2)_2$ (aka DTN) undergoes a spin-gap closure transition at about 4.2 kbar, resulting in a pressure-induced magnetic ordering. The studies are complemented by high-pressure-electron spin-resonance measurements confirming the proposed scenario. Powder neutron diffraction measurements revealed that no lattice distortion occurs at this pressure and the high spin symmetry is preserved, establishing DTN as a perfect platform to investigate $z = 1$ quantum critical phenomena. The experimental observations are supported by DMRG calculations, allowing us to quantitatively describe the pressure-driven evolution of critical fields and spin-Hamiltonian parameters in DTN.

Magnetic insulators with their short-range interactions and well-controlled effective Hamiltonians offer an ideal playground for studying quantum critical phenomena induced by various stimuli[1–3]. Nowadays, great attention is attracted by antiferromagnetic (AF) ordering, which can be induced, e.g., by magnetic field in gapped quantum magnets[4,5]. On the other hand, especially tantalizing is the possibility of magnetic ordering, accompanied by spontaneous spin-symmetry break-down without the aid of magnetic field, while keeping the spectrum degenerate in the absence of Zeeman splitting[6]. Such

transitions are known to appear in certain quantum antiferromagnets under applied pressure. As Fig. 1a illustrates, both the magnetic field and pressure can be regarded as a tuning parameter. Contrary to the field-induced transitions, the pressure-induced case belongs to another class of universality[7–9]. The key distinction is the dynamic critical exponent $z$ that links the characteristic energy of spin excitations to the momentum relative to the critical one:

$$\omega \propto |q - q_c|^z. \tag{1}$$

[1]Dresden High Magnetic Field Laboratory (HLD-EMFL) and Würzburg-Dresden Cluster of Excellence ct.qmat, Helmholtz-Zentrum Dresden-Rossendorf (HZDR), Dresden, Germany. [2]National High Magnetic Field Laboratory, Tallahassee, FL, USA. [3]Institut für Festkörper- und Materialphysik, Technische Universität Dresden, Dresden, Germany. [4]Research Facility Center for Science and Technology, Kobe University, Kobe, Japan. [5]Molecular Photoscience Research Center, Kobe University, Kobe, Japan. [6]Graduate School of Science, Kobe University, Kobe, Japan. [7]Institute for Materials Research, Tohoku University, Sendai, Japan. [8]Neutron Scattering Division, Oak Ridge National Laboratory, Oak Ridge, TN, USA. [9]Laboratory for Solid State Physics, ETH Zürich, Switzerland. [10]Instituto de Fisica, Universidade de São Paulo, São Paulo, Brazil. [11]Max Planck Institute for Chemical Physics of Solids, Dresden, Germany. ✉e-mail: k.povarov@hzdr.de; s.zvyagin@hzdr.de

**a**

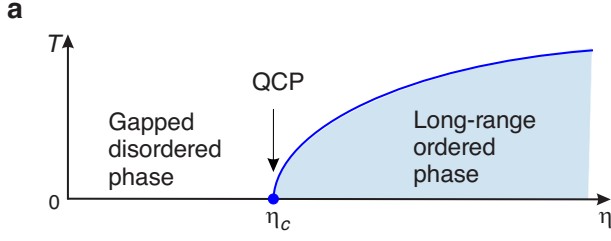

**b**

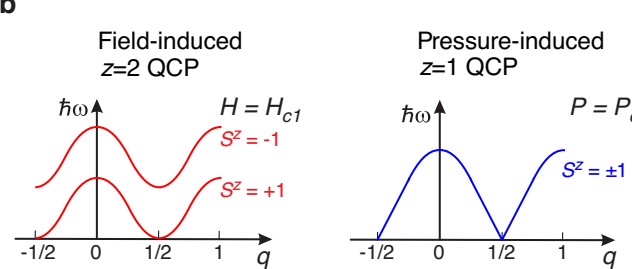

Fig. 1 | **Quantum phase transitions in a spin-1 AF with large planar anisotropy.**
**a** The generic phase diagram ($\eta$ is the tuning parameter; QCP is the quantum critical point). The critical value of the tuning parameter is labeled $\eta_c$. **b** Schematic view of magnetic excitation spectra for the field- and pressure-induced quantum phase transitions ($q$ is the momentum; $\hbar\omega$ is the excitation energy; $z$ is the dynamic critical exponent, see Eq. (1)).

Field-induced phase transitions are characterized by $z = 2$ and a quadratic low-energy excitation spectrum, while the pressure-induced case has $z = 1$ and a linear spectrum[6,8]. The realizations of these two scenarios are illustrated in Fig. 1b. The $z = 1$ regime is very remarkable. It should have mean-field critical exponents[8] and the universal scaling of dynamic fluctuations at the same time[10]. This quantum critical point (QCP) is also a natural habitat for a well-defined order-parameter amplitude mode[11,12].

There is a number of gapped quantum antiferromagnets, demonstrating field-induced AF ordering with $z = 2$[5]. If the spin Hamiltonian of a system has axial symmetry with respect to the applied field, the field-induced AF ordering can be formally described as the Bose−Einstein condensation of magnons by mapping the spin-1 system into a gas of semi-hard-core bosons[4,13]. In order to perform an accurate comparison with the theory, high-symmetry spin systems are highly demanded. The compound $NiCl_2 \cdot 4SC(NH_2)_2$ (dichloro−tetrakis thiourea−nickel(II), known as DTN) is one of them, having a tetragonal crystal structure (space group $I4$) and easily accessible critical fields.

A good realization of the true $z = 1$ QCP is actually hard to find. Most of the materials known to show such a transition ($TlCuCl_3$[11,12], $KCuCl_3$[14,15], $(C_4H_{12}N_2)Cu_2Cl_6$[16], and $(C_9H_{18}N_2)CuBr_4$[17]) suffer from unwanted anisotropies. For $TlCuCl_3$ and $(C_9H_{18}N_2)CuBr_4$ biaxial anisotropy was experimentally detected[18,19]. In $KCuCl_3$ the presence of biaxial anisotropy follows from symmetry considerations, and is indirectly evidenced by a peculiar orientation of sublattice magnetization in the ordered phase[14]. Such anisotropies eventually lead to the criticality of the Ising universality class[20], different from the target QCP. This does not seem to be the case for $CsFeCl_3$[21,22], but there, one has to deal with geometric exchange frustration.

In the present study, utilizing high-pressure tunneling-diode oscillator (TDO) susceptibility, ultrasound-propagation measurements, and high-field electron spin resonance (ESR) techniques, we demonstrate a pressure-induced phase transition in DTN, which we ascribe to the long-sought $z = 1$ criticality. This transition resides at an easily accessible pressure of about 4.2 kbar. Neutron-diffraction measurements confirm the absence of a structural transition and reveal an undistorted tetragonal symmetry near this QCP. At higher pressure, we

actually find an irreversible distortion of the lattice occurring. We describe the experimentally measured phase boundaries employing a quasi-1D numerical approximation, circumventing a renormalization of the spin-Hamiltonian parameters by quantum fluctuations.

## Results

### Magnetism in DTN: brief introduction

Magnetism in DTN originates from the spin-1 $Ni^{2+}$ ions forming a tetragonal lattice (we refer the reader to Supplemental Material S1 for detailed crystallographic information). The magnetic properties are defined by the competition between the strong single-ion planar anisotropy $D$, and the antiferromagnetic exchanges interactions $J_c$ and $J_a \equiv J_b$ along the corresponding $c$ and $a, b$ directions. The effective Hamiltonian is

$$\hat{H} = \sum_{\mathbf{r}} D(\hat{S}_{\mathbf{r}}^z)^2 + J_c \hat{\mathbf{S}}_{\mathbf{r}} \hat{\mathbf{S}}_{\mathbf{r}+\mathbf{c}} + J_a(\hat{\mathbf{S}}_{\mathbf{r}} \hat{\mathbf{S}}_{\mathbf{r}+\mathbf{a}} + \hat{\mathbf{S}}_{\mathbf{r}} \hat{\mathbf{S}}_{\mathbf{r}+\mathbf{b}}). \quad (2)$$

Here, $\mathbf{r}$ runs along the nickel positions in the tetragonal sublattice; $\mathbf{a}$, $\mathbf{b}$, and $\mathbf{c}$ are the primitive lattice translation vectors towards the neighboring spins. Importantly, the symmetry prohibits any in-plane second-order anisotropic terms. The planar anisotropy protects the spin-singlet state $|S^z = 0\rangle$ on every magnetic ion, thus preventing Néel order in zero field, otherwise favored by the exchange interactions. The set of constants describing DTN at ambient pressure was initially obtained from zero-field neutron spectroscopy and thermodynamic studies[23,24], and later refined as $D/k_B = 8.9$ K, $J_c/k_B = 1.82$ K, and $J_a/k_B = 0.34$ K[25] based on the spectroscopic properties at high magnetic fields (where the effect of quantum renormalization is not present). The zero-field gap $\Delta/k_B \simeq 3.5$ K[26] is relatively small compared to the excitation-doublet bandwidth of ~ 8 K. If the magnetic field is applied along the $c$ ($z$) direction, the degeneracy of the doublet is removed due to Zeeman splitting (as shown in Fig. 1b), triggering eventually the onset of low-temperature magnetic order at $\mu_0 H_{c1} \simeq 2.1$ T[23,27]. At stronger magnetic field, $\mu_0 H_{c2} \simeq 12.2$ T, DTN undergoes the transition into the fully spin-polarized state. We would like to stress that the zero-field gap in DTN is dominantly determined by the single-ion anisotropy (in contrast to Haldane spin-chain materials), making DTN a rare example of a large-$D$ spin-1 system.

### High-pressure neutron diffraction

As the pressure increases, the parameters of Hamiltonian Eq. (2) are expected to change. However, it is necessary to check first whether the lattice remains undistorted. To this end, we have performed a series of structural neutron diffraction studies[28]. As shown in Fig. 2, the DTN lattice is smoothly compressible up to about 6 kbar. Interestingly, the compression goes in a rather uniaxial manner, with shrinking of the sample mostly along the $c$ axis ($\partial c/\partial P = -0.038 \pm 0.003$ Å/kbar at 1.8 K). This implies the compression of the main Ni−Cl−Cl−Ni superexchange pathway and the deformation of the $Ni^{2+}$ local environment. Such changes must affect $D$ and $J_c$ in the first place. The lattice parameter $a$ remains nearly constant with $\partial a/\partial P = (-4.1 \pm 1.5) \cdot 10^{-3}$ Å/kbar. At $P_{irr} \simeq 6$ kbar, an irreversible structural transition occurs, evidenced by discontinuities in the lattice parameters. A more detailed discussion is given in the Supplemental Material (S2 and S3).

### High-pressure TDO measurements

The TDO susceptibility technique is well established as a versatile tool for detecting field-induced phase transitions in solids under applied pressure[29−32].

The results of our measurements are shown in Fig. 3. At low pressures, two anomalies, corresponding to the boundaries of the long-range AF ordered phase at $H_{c1}$ and $H_{c2}$, respectively, are well discernible. To extract the critical fields in a reliable way we use a set of empirical functions (red lines, a detailed description is given in the

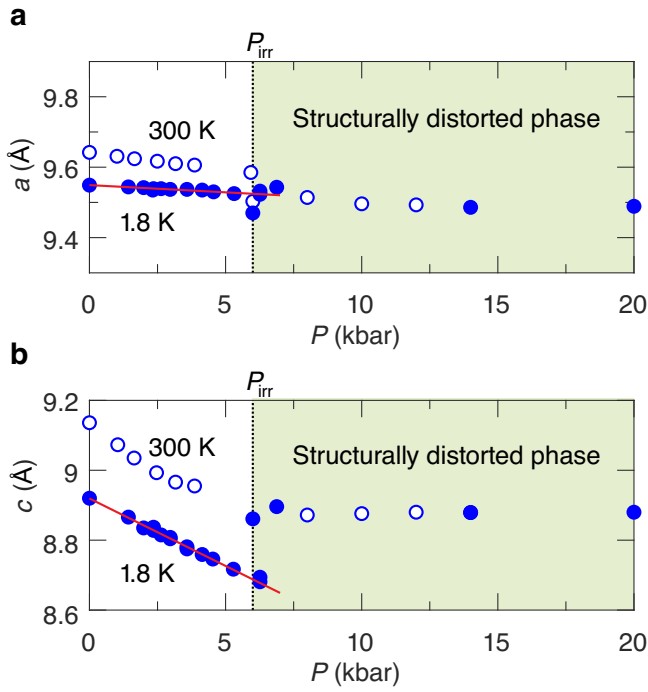

**Fig. 2 | Pressure dependence of lattice parameters of DTN, as measured by neutron diffraction. a** Pressure dependence of the lattice parameter $a$. **b** Pressure dependence of the lattice parameter $c$. Open and closed symbols correspond to 300 and 1.8 K data, respectively. Dotted lines denote the irreversible structural phase transition at $P_{irr} \sim 6$ kbar. Red lines are fit results (see text for details).

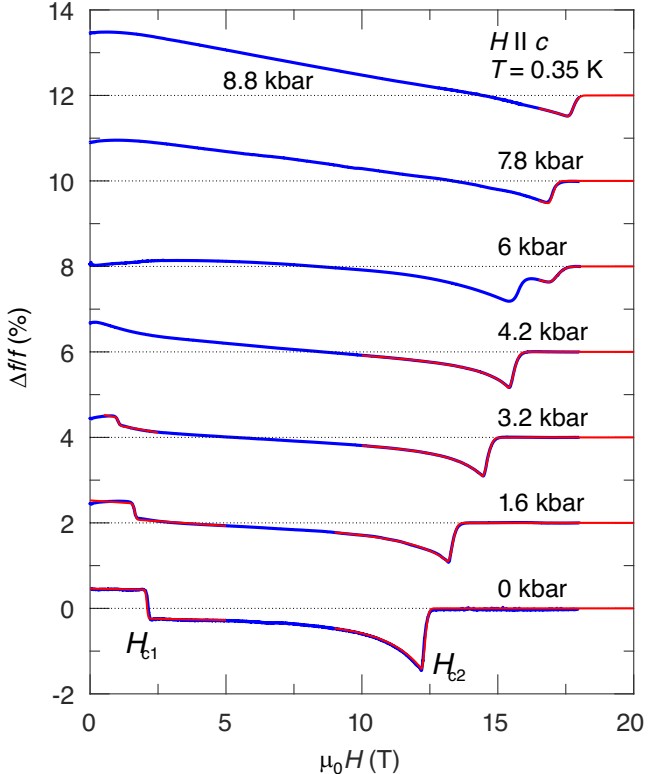

**Fig. 3 | Pressure dependence of the TDO frequency changes $\Delta f/f$ in response to the magnetic field (the data are offset for clarity).** Results of empirical fits are shown in red (see Supplemental Material for details).

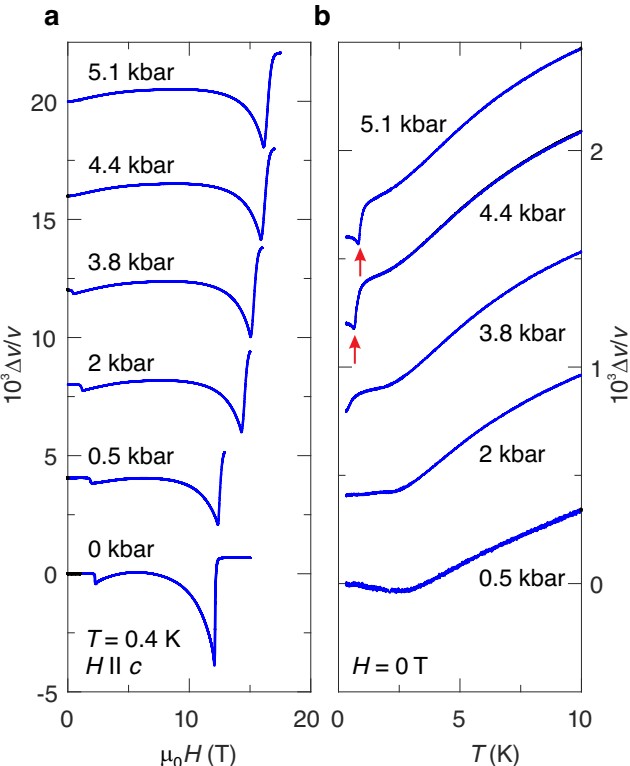

**Fig. 4 | Pressure dependences of ultrasonic properties of DTN. a** Relative change of the sound velocity $\Delta v/v$ of the longitudinal acoustic mode as function of the magnetic field (zero-pressure data are taken from ref. 33 and were measured at 0.3 K). The data are offset for clarity. **b** Relative change of the sound velocity $\Delta v/v$ of the longitudinal acoustic mode as function of temperature. The data are offset for clarity. The onset of magnetic order is marked by red arrows.

Supplemental Material). With applied pressure, we observe a decrease and increase of the first and second critical field, respectively. Above 4.2 kbar, the low-field transition is not visible anymore. This is a strong evidence of the spin-gap closure. The critical field $H_{c2}$ increases linearly up to $P_{irr}$, where a discontinuity appears. The high-field part of the curve at 6 kbar (about $P_{irr}$) features a double-dip structure. This reflects the coexistence of two structurally different phases at the first-order crystallographic transition. In the structurally distorted phase above $P_{irr}$, we observe only one anomaly.

### High-pressure ultrasound measurements

In order to reveal the nature of the pressure-induced gapless phase, we performed high-pressure ultrasound measurements. The magnetic ordering in DTN at zero pressure was thoroughly investigated with ultrasound by Chiatti et al.[33,34], firmly establishing the connection between the long-range-order onset and the sound velocity anomalies in DTN. These anomalies reflect the spin-susceptibility divergence in the vicinity of the critical temperature[35], as a result of the pronounced magnetoelastic coupling in DTN.

We studied the longitudinal sound mode along the $c$ direction (mode $c_{33}$) at high pressures. The field-induced changes of its velocity are shown in Fig. 4a. Similar to our TDO data (Fig. 3), there are two anomalies at low pressures. As revealed previously[33,34], they evidence the transitions from disordered gapped to AF ordered gapless state at $H_{c1}$, and the subsequent transition to the fully spin-polarized state at $H_{c2}$.

At higher pressures (above 3.8 kbar), we only observe the high-field phase transition, while the gapless AF ordered phase

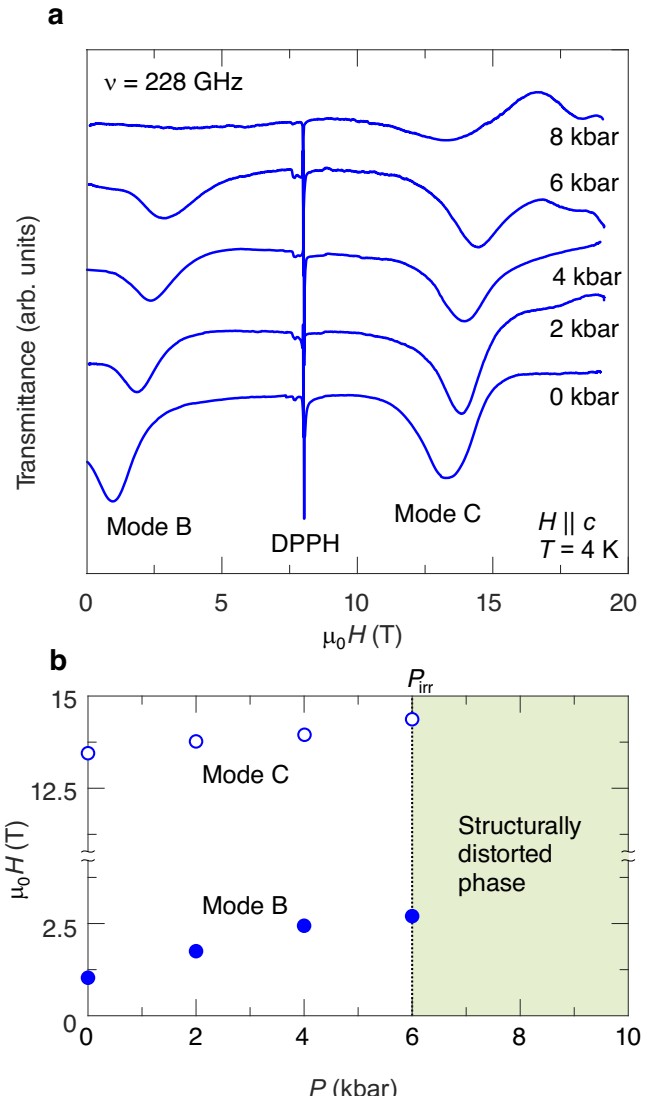

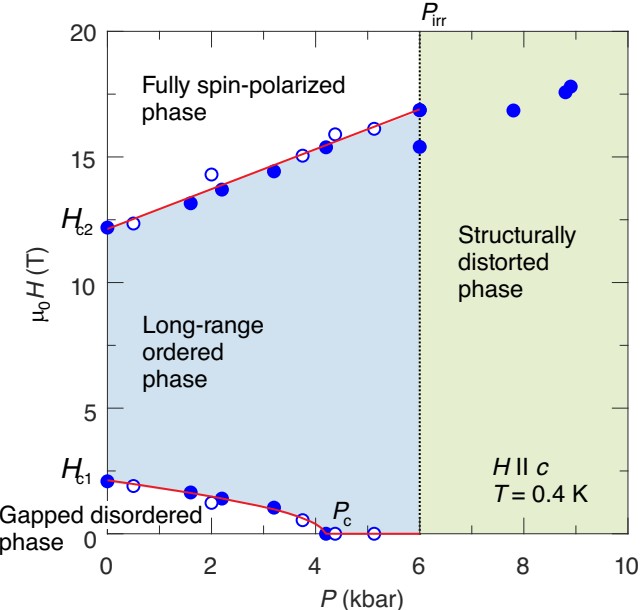

**Fig. 6 | Field-pressure phase diagram of DTN.** Open and closed symbols correspond to ultrasound and TDO data, respectively. Red lines are model fits (see the text for details). The dotted line corresponds to the irreversible structural phase transition at $P_{irr} \sim 6$ kbar.

## Discussion

The results of our TDO and ultrasound experiments on DTN are summarized in the phase diagram in Fig. 6. The data ensure that near 4 kbar the spin gap does close and antiferromagnetic order emerges. One can theoretically describe the pressure-dependent spin Hamiltonian, consistent with these observations. The saturation field $H_{c2}$ is known to follow the linear spin wave theory (LSWT) description without any quantum renormalization[25]:

$$g\mu_B\mu_0 H_{c2} = D + 4J_c + 8J_a. \qquad (3)$$

Given the overall smallness of $J_a$ and lack of a significant pressure-induced length change in this direction, we can neglect possible variations in this interaction. Thus, the phase-diagram evolution is due to the changes in $D$ and $J_c$:

$$g\mu_B\mu_0 \frac{\partial H_{c2}}{\partial P} = \frac{\partial D}{\partial P} + 4\frac{\partial J_c}{\partial P}. \qquad (4)$$

With the g-factor being 2.26, the measured slope $\mu_0 \partial H_{c2}/\partial P = 0.78 \pm 0.03$ T/kbar (upper red line in Fig. 6 below the instability pressure $P_{irr}$) quantifies the linear pressure dependence of $D$ and $J_c$.

The analysis of the first critical field poses a greater challenge: since $g\mu_0\mu_B H_{c1} = \Delta$, it requires evaluating the zero-field spin gap for the given set of Hamiltonian (2) parameters. No exact theory is available for that to the best of our knowledge. We overcome this challenge employing a random-phase approximation (RPA)[37] based ansatz combined with density matrix renormalization group (DMRG) calculations[38,39]. The starting point is the determination of the zero-field gap value $\Delta_0(D/J_c)$ in the $J_a = 0$ limit of a single anisotropic chain, utilizing the latter technique. As far as we know, only a particular range close to the quantum critical point at $D/J_c \sim 1$ was thoroughly accessed with DMRG earlier[40]. The next step of the model is to apply the RPA treatment to interacting chains in order to estimate the critical value of the coupling $J_a^{crit}$ that closes the gap. The details of the calculations

---

**Fig. 5 | High-field ESR results. a** Selected ESR spectra taken at a frequency of 228 GHz ($H \| c$, $T = 4$ K) at different pressures [the sharp line corresponds to DPPH (2,2diphenyl-1-picrylhydrazyl), used as a marker]. **b** Pressure dependencies of ESR fields for modes B and C, as revealed by experiment.

becomes extended to zero magnetic field. An additional evidence for the pressure-induced magnetic ordering in zero fields is our observation of the sound–velocity anomaly below 1 K (red arrows in Fig. 4b). As expected, the ordering temperature grows with increasing pressure.

### Electron spin resonance

The applied pressure in DTN should affect not only the ground state but the spin dynamics as well. Electron spin resonance (ESR) has recently proven to be a powerful tool to probe the excitation spectra in strongly correlated spin systems under applied pressure[29,36]. Selected examples of ESR spectra in DTN are shown in Fig. 5a (adopting the naming convention from ref. 25, we label the observed modes B and C). Excitation modes corresponding to $\Delta S^z = 1$ transitions remain well visible up to 6 kbar. With the pressure increase both modes demonstrate a gradual shift towards higher fields (as illustrated in Fig. 5b) until they vanish in the structurally distorted phase above about 6 kbar. The complete frequency-field diagrams can be found in the Supplemental Material S6.

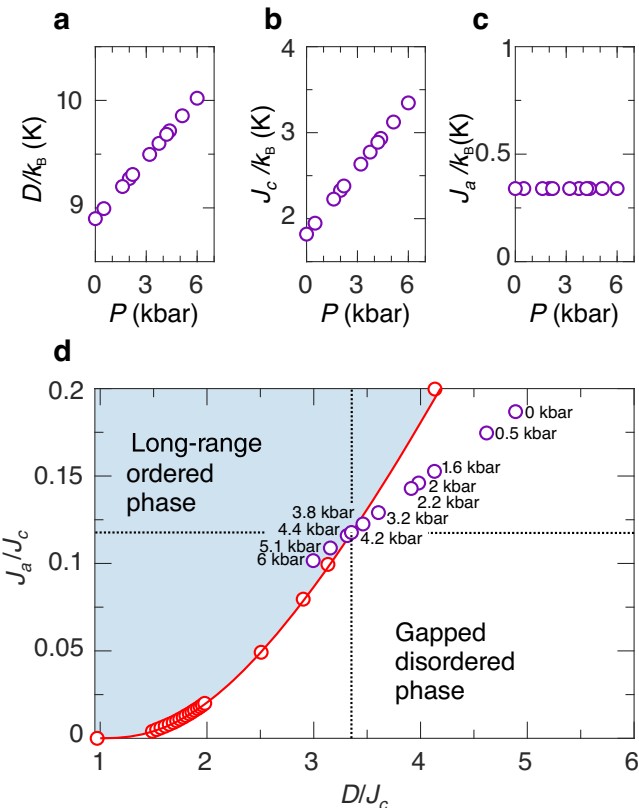

**Fig. 7 | Results of model calculations. a** Pressure dependence of the anisotropy parameter $D/k_B$. **b** Pressure dependence of the exchange coupling parameter $J_c/k_B$. **c** Pressure dependence of the transverse exchange coupling parameter $J_a/k_B$. The calculation results are shown for the pressures, matching the experiments. **d** The Sakai–Takahashi phase diagram for a quasi-1D spin-1 antiferromagnet[44]. The red open points correspond to QMC calculations for the phase boundary[45]; the red line is a prediction based on the ansatz as described in the text. The purple circles correspond to estimates of the spin Hamiltonian parameters at given pressures. The dotted cross marks the critical parameter values at 4.2 kbar.

(based on large-$D$ approximation for the dynamic structure factor[41–43] and the Kramers–Kronig relations) are given in the Supplemental Material (S4).

The main result is the approximation for the order-disorder zero-field phase boundary:

$$J_a^{crit} = \frac{\Delta_0^2}{4AD}. \tag{5}$$

We find that the numerical value $1/4A \simeq 0.14$ provides an excellent description of this phase boundary, previously obtained by extensive quantum Monte–Carlo (QMC) simulations[44,45]. This comparison can be seen in Fig. 7d, where the calculation result using Eq. (5) is shown as the red line. Then, it follows from our approach that the value of the gap, renormalized by $J_a$, is

$$\Delta = \Delta_0 \sqrt{1 - \frac{J_a}{J_a^{crit}}}. \tag{6}$$

Hence, the first critical field can be expressed through the pressure-dependent Hamiltonian parameters as

$$g\mu_0\mu_B H_{c1}(P) = \Delta\left[\Delta_0\left(\frac{D(P)}{J_c(P)}\right), D(P)\right]. \tag{7}$$

The ambient pressure $H_{c1}$ evaluated by this procedure agrees with the experimental value, ensuring the validity of the approach for the smaller gap values as well.

The ability to predict both critical fields using Eqs. (3) and (7) finally opens a route to a self-consistent treatment of the measured phase diagram without any "effective" parameters. We numerically optimize the agreement between (7) and the experimental $H_{c1}$ data by varying the choice of the critical pressure $P_c$ at which the gap closes completely. This fixes a unique combination of $\partial(D, J_c)/\partial P$. The optimization results in the critical pressure $P_c = 4.2 \pm 0.3$ kbar (see Supplemental Material S5 for more details). The magnetic phase diagram (Fig. 6) is fully captured with $k_B^{-1}\partial D/\partial P = 0.16 \pm 0.03$ K/kbar and $k_B^{-1}\partial J_c/\partial P = 0.25 \pm 0.01$ K/kbar. Figure 7a–c illustrate how the spin-Hamiltonian parameters of DTN are pressure-tuned up to $P_{irr}$. In the phase diagram of Fig. 7d, we show how the corresponding ground state (illustrated by purple points) is changing from gapped to long-range ordered at $P_c$ in accordance with the previous discussion.

The rich high-pressure physics of DTN under pressure invites further investigations, related to the interplay of quantum and thermal fluctuations near $z = 1$ quantum criticality[9,12,46–48]. In particular, it would be important to access the order parameter (e.g., with neutron diffraction or nuclear magnetic resonance) and the extended field-temperature-pressure phase diagram[32]. This would open the possibility to establish a direct connection between the minimalistic and highly symmetric spin Hamiltonian of DTN and the effective field theory used to describe the static and dynamic properties of critical quantum magnets[49,50]. The undistorted symmetry also makes DTN a perfect candidate for investigating of the effect of thermal and quantum fluctuations on the amplitude mode[12] in the absence of Ising-type anisotropy. One can also expect chemically substituted DTNX[51,52] to be a fruitful playground for the interplay of quantum $z = 1$ criticality and quenched disorder, similarly to $(C_4H_{12}N_2)Cu_2(Cl_{1-x}Br_x)_6$[53], $(NH_4)_xK_{1-x}CuCl_3$[54], and $Cs_{1-x}Rb_xFeCl_3$[55]. The higher symmetry and a simpler, well-understood Hamiltonian make the case of DTNX much more appealing for such studies.

To summarize, we have identified the material DTN as a unique platform for studies of the exotic $z = 1$ universality class in a three-dimensional magnetic material. By means of TDO, ultrasound, and ESR measurements we have confirmed the existence of pressure-induced criticality, separating gapped disordered and gapless long-range ordered magnetic phases in the material. The nearly ideal axial symmetry of the structure is retained at that point, which makes DTN to stand out among the non-frustrated quantum magnets. In addition to that, we have extracted the pressure dependence of the spin Hamiltonian parameters from the data, and have achieved a quantitative theoretical understanding of the transition mechanism.

## Methods
### Sample growth
Samples for the thermodynamic measurements and the electron spin resonance spectroscopy were synthesized in the University of São Paulo from aqueous solution using a thermal-gradient method[56,57]. Samples used in the neutron diffraction studies were synthesized at ETH Zürich. Fully deuterated chemicals (water and thiourea) were used. The crystals were crushed into powder for these experiments.

### High-pressure neutron diffraction
Neutron-diffraction experiments with the powder samples were performed at instrument HB2a at the High Flux Isotope Reactor, Oak Ridge National Laboratory (Oak Ridge, Tennessee, USA). Up to a few grams of deuterated DTN powder material were used. A Ge[113] or Ge[115] vertically focussing wafer-stack monochromator was used to produce a neutron beam with 2.41 Å or 1.54 Å wavelength, respectively. A $^4$He Orange cryostat with an aluminum He-gas pressure cell (capable

of producing pressures up to 6 kbar) and a CuBe clamp cell (capable of producing pressures up to 20 kbar) were used. The pressure in the gas cell was measured in situ using a manometer. For the clamp cell a rock salt (halite) calibration curve was used for the pressure estimate, resulting in an error bar of about 1 kbar. After initial data reduction, the `FULLPROF` package[58] was used for the intensity profile analysis and structure determination. More details of these experiments can be found in ref. [28].

## High-pressure TDO

High-pressure TDO measurements were conducted at the National High Magnetic Field Laboratory, Florida State University (Tallahassee, Florida, USA) in magnetic fields up to 18 T using a TDO susceptometer[59]. The magnetic field was applied along the $c$ axis of the crystal placed in a tiny copper-wire coil with 0.8 mm diameter and 1 mm height. This assembly was immersed into Daphne 7575 oil and encapsulated in a Teflon cup inside the bore of a piston-cylinder pressure cell made of a chromium alloy (MP35N). The pressure created in the cell was calibrated at room temperature and again at low temperature using the fluorescence of the $R1$ peak of a small ruby chip as a pressure marker[60] with an accuracy better than 0.15 kbar. The pressure cell was immersed directly into $^3He$, allowing TDO measurements down to 350 mK.

## High-pressure ultrasound measurements

We performed ultrasound measurements using the pulse-echo method with phase-sensitive detection technique[35,61]. Overtone polished $LiNbO_3$ transducers with 36° Y-cut (longitudinal sound polarization) were used to generate and detect ultrasonic signals with a frequency of about 35 MHz. The transducers were bonded to natural (001) crystal surfaces of a DTN sample with Thiokol 32, providing $k\|u\|H\|c$ experiment geometry. The crystal dimensions were $0.3 \times 0.3 \times 2.95$ mm$^3$. We determined the low-$T$ sound velocity at zero field and pressure to $v \simeq 2600$ m/s, in agreement with[33,34]. A commercially available piston-cylinder cell with CuBe outer sleeve, NiCrAl inner sleeve, and tungsten carbide inner pistons (C&T Factory Co.,Ltd) was adapted for ultrasound experiments following [62], with Daphne oil 7373 as a pressure medium. The pressure cell (with a calibrated $RuO_2$ temperature sensor on the outer side) was thermally anchored to the $^3He$ pot of the cryostat via a copper rod. The pressure was estimated from the pressure-dependent superconducting transition of tin[63,64].

## High-pressure ESR

High-pressure ESR studies of DTN were performed employing a 25 T cryogen-free superconducting magnet ESR setup (25T-CSM) at the High Field Laboratory for Superconducting Materials, Institute for Materials Research, Tohoku University (Sendai, Japan)[65,66]. Gunn diodes were utilized as microwave sources for frequencies up to 405 GHz; the transmitted radiation power was detected using a hot-electron InSb bolometer operated at 4.2 K. A DTN crystal was loaded into a Teflon cup filled with Daphne 7474 oil as a pressure medium. A two-section piston-cylinder pressure cell made from NiCrAl (inner cylinder) and CuBe (outer sleeve) has been used. The key feature of the pressure cell is the inner pistons, made of $ZrO_2$ ceramics. The applied pressure was calibrated against the superconducting transition temperature of tin[63,64], detected by AC susceptometer. The actual pressure during the experiment was calculated using the relation between the load at room temperature and the pressure obtained at around 3 K; the pressure calibration accuracy was better than 0.5 kbar[67].

## DMRG calculations

The DMRG calculations utilized the `Julia` version of the `ITensors` package[68,69]. A 249-site anisotropic $S = 1$ chain was simulated. The calculations were performed on the `hemera` cluster (HZDR).

## Data availability

The data that support the findings of this study are available from the corresponding author upon request.

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

## Acknowledgements

We thank A. Mannig, J. Möller, and G. Perren for their involvement at the early stage of the ETH Zürich part of the project. This work was supported by the Deutsche Forschungsgemeinschaft through ZV 6/2-2, the Würzburg-Dresden Cluster of Excellence on Complexity and Topology in Quantum Matter - *ct.qmat* (EXC 2147, Project No. 390858490) and the

SFB 1143 (Project No. 247310070), as well as by HLD at HZDR, member of the European Magnetic Field Laboratory (EMFL) [K.Yu.P., S.A.Zv., S.Zh., A.H., J.W.]. The neutron-diffraction experiments at HB2a used resources at the High Flux Isotope Reactor, a Department of Energy Office of Science User Facility operated by the Oak Ridge National Laboratory. This work was partially supported by the Swiss National Science Foundation, Division II [A.Z.]. A portion of this work has been performed at the National High Magnetic Field Laboratory, which is supported by the National Science Foundation Cooperative Agreement DMR-1644779 and the State of Florida [D.E.G.]. ESR experiments were performed at the High Field Laboratory for Superconducting Materials, Institute for Materials Research, Tohoku University (proposal 19H0501 and 20H0501). Support of the ICC-IMR Visitor Program at Tohoku University is acknowledged [S.A.Zv.]. This work was partially supported by the Brazilian agencies CNPq (grant 304455-2021-0) and FAPESP (grant 2021-12470-8) [A.P.F.]. We also thank M. E. Zhitomirsky for fruitful discussions.

## Author contributions

S.A.Zv. conceived the project and performed the TDO (together with D.E.G.) and ESR (together with T.S., S.K. and H.N.) measurements; K.Yu.P. performed the analysis of this data and the numeric simulations. V.O.G. has performed the neutron diffraction measurements and analyzed the data. A.H. and S.Zh. have performed the ultrasound propagation measurements. M.N. has contributed to the preliminary high-pressure thermodynamic measurements. A.P.F. has synthesized the non-deuterated single crystals used in the study. H.O., H.N., A.Z. and J.W. administered at work at Kobe University and Tohoku University, ETH Zürich, and HZDR, respectively. All the authors have contributed to the discussions and manuscript writing.

## Funding

## Competing interests

The authors declare no competing interests.
