## [Peer Review File · Nature Communications]

REVIEWER COMMENTS

Reviewer #1 (Remarks to the Author):

The organic compound $\text{NiCl}_2 \cdot 4\text{SC}(\text{NH}_2)_2$ (DTN) is a gapped $S=1$ system with a single-ion anisotropy dominating over the exchange coupling. DTN represents one of the finest experimental manifestations of a Bose-Einstein condensation (BEC) phase with the dynamic critical exponent $z=2$ at the magnetic field-induced quantum critical point and was extensively investigated in this context. The work by Povarov et al., reports the experimental realization of the pressure-induced quantum phase transition with $z=1$ in DTN. The phase diagram as a function of pressure and magnetic field were mapped out by the TDO susceptibility measurements and the transition to a gapless phase above 4 kbar is evidenced by disappearance of the lower critical field. The corresponding phase boundaries can be quantitatively described by the DMRG plus RPA calculations. Moreover, the ESR spectra under pressure can be fairly reproduced by the GSWT calculations using the Hamiltonian parameters obtained at different pressures.

In my perspective, the manuscript is presented well, their analysis appears to be sound and logical, and the conclusions are well-supported. So far, there are very limited experimental studies on $z=1$ quantum critical points due to a lack of experimental systems. In this regard, I think this work will serve as a valuable addition to the list of the materials that showcase quantum phase transition with $z=1$ universality class and

foster advancements in this field. Therefore, I would recommend it for publication in Nature Communications after the authors address my following comments:

- a. Referring to an “Higgs mode” in an antiferromagnet without local gauge invariance appears to be misleading. The term “the amplitude or longitudinal mode” is more appropriate and descriptive for it.
- b. Note that TiCuCl_3 has a XY-type exchange anisotropy (e.g., Phys. Rev. Lett. 100, 205701 (2008)). There seems no magnetic anisotropy for $(\text{C}_4\text{H}_{12}\text{N}_2)\text{Cu}_2\text{Cl}_6$ and KCuCl_3 .
- c. Figure 2c shows the change of lattice parameters with pressure. The results obtained in a gas and clamp pressure cell do not seem to align with each other. Considering that the pressure in the gas pressure cell was determined in situ via the manometer, it is very likely that the actual pressure deviates from the measured value. To prevent any confusion, I would suggest disregarding the results obtained in the gas pressure cell.
- d. To support their conclusion (i.e., the onset of the antiferromagnetic order induced by the pressure) more robustly, can the authors consider further supporting measurements, such as neutron diffraction, or muon spin relaxation, or heat capacity with He-3 or dilution refrigerator?

Reviewer #2 (Remarks to the Author):

The authors investigated the pressure-driven gapped-to-gapless phase transition in $\text{NiCl}_2 \cdot 4\text{SC}(\text{NH}_2)_2$ (DTN). They suggested an occurrence of antiferromagnetic (AFM) order induced by pressure, and quantitatively described the pressure-driven evolution of the critical fields and spin Hamiltonian parameters in DTN. They investigated the DNT under pressure by means such as tunnel diode oscillator

(TDO) susceptibility measurements, electronic spin resonance (ESR) spectra, neutron diffraction studies. However, DTN, which is an old system, has been intensively investigated previously. Thus, it is lack of broad interests. Besides, there are some other problems with this manuscript.

1. The authors declared that the quantum-disordered state (QD) is suppressed by the pressure, and an AFM ordering is induced. However, the experiment just shows the suppression of QD, but not the emergence of AFM. As we know, at the quantum critical point, the suppression of a phase sometimes is sometimes accompanied by the appearance of another new phase. We admit the suppression of the QD. However, the phase above P_c need to be determined very carefully. Therefore, how to undoubtedly prove it is an AFM phase? And what is the magnetic state above 6 kbar?

2. In the Figure 6 of the ESR spectra, there is no obvious change before and after the critical pressure P_c . The ESR is performed at $T=4K$. It seems that this measurement has no relation with the QCP discussed here.

3. In Figure 3 of TDO, the H_{c1} disappears between 3.2 kbar and 4.2kbar, while the H_{c2} persists through the pressure-induced structural phase transition. Only at 6.0 kbar, the H_{c2} splits into two coexisted peaks. what does the coexistence mean? Since TDO is corresponding to the magnetization, does it mean that the coexistence of two kinds of magnetic ordering? Furthermore, what are exactly these two magnetic ordering phases?

4. Due to the deficiency of spectra between 3.2 kbar and 4.2kbar in Figure 3, it is hard to determine the P_c from experiment within 1 kbar.

5. This DNT exhibits an irreversible distortion of the lattice at higher pressures exceeds 6 kbar. Is it reversible or irreversible below 6 kbar? Since the irreversible distortion, these experimental should be performed on different samples. Thus, how to guarantee the same quality of these samples used?

6. In Figure S4, RPA ansatz fit gives that $P_c= 4.3$ kbar. However, in figure 3, at 4.2 kbar, the H_{c1} has been disappeared. What fact caused this large deviation of P_c ?

Summary, the author investigated the DTN under pressure by using several challenging measurement techniques, such as TDO, ESR spectra, neutron diffraction under pressures. These devices are very excellent in the area of experimental study. However, it seems that only one experimental measurement is related to the QCP discussed. Moreover, this measurement cannot give the solid proof for the new phase above the P_c . Due to these reasons, I cannot recommend the publication of this article in this journal.

Reply to Referee 1

The organic compound $\text{NiCl}_2 \cdot 4\text{SC}(\text{NH}_2)_2$ (DTN) is a gapped $S = 1$ system with a single-ion anisotropy dominating over the exchange coupling. DTN represents one of the finest experimental manifestations of a Bose-Einstein condensation (BEC) phase with the dynamic critical exponent $z = 2$ at the magnetic field-induced quantum critical point and was extensively investigated in this context. The work by Povarov *et al.*, reports the experimental realization of the pressure-induced quantum phase transition with $z = 1$ in DTN. The phase diagram as a function of pressure and magnetic field were mapped out by the TDO susceptibility measurements and the transition to a gapless phase above 4 kbar is evidenced by disappearance of the lower critical field. The corresponding phase boundaries can be quantitatively described by the DMRG plus RPA calculations. Moreover, the ESR spectra under pressure can be fairly reproduced by the GSWT calculations using the Hamiltonian parameters obtained at different pressures.

In my perspective, the manuscript is presented well, their analysis appears to be sound and logical, and the conclusions are well-supported. So far, there are very limited experimental studies on $z = 1$ quantum critical points due to a lack of experimental systems. In this regard, I think this work will serve as a valuable addition to the list of the materials that showcase quantum phase transition with $z = 1$ universality class and foster advancements in this field. Therefore, I would recommend it for publication in Nature Communications after the authors address my following comments.

We would like to thank the Referee for the very positive comments and deep assessment of our manuscript. We especially appreciate the recognition of importance of the present results indicating DTN as a perfect candidate with $z = 1$ universality class.

Below we address the comments and suggestions of the Referee:

a. Referring to an “Higgs mode” in an antiferromagnet without local gauge invariance appears to be misleading. The term “the amplitude or longitudinal mode” is more appropriate and descriptive for it.

We fully agree with this comment. The “Higgs” notation was mentioned there simply because such terminology has occurred in the literature before. In the revised manuscript, we erased the “Higgs” notation and left only the “order parameter amplitude mode” notation in the text.

b. Note that TlCuCl_3 has a XY-type exchange anisotropy (e.g., Phys. Rev. Lett. 100, 205701 (2008)). There seems no magnetic anisotropy for $(\text{C}_4\text{H}_{12}\text{N}_2)\text{Cu}_2\text{Cl}_6$ and KCuCl_3 .

We have put a bit more emphasis on the anisotropy issues of the aforementioned materials. In the neutron investigation by Rüegg *et al.*, an XY anisotropy is already sufficient for the data description. However, more detailed ESR investigations [Glazkov *et al.*, Phys. Rev. B 69, 184410 (2004)] demonstrate that the anisotropy actually has biaxial character.

Similarly, low-frequency ESR unambiguously detects biaxial anisotropy in PHCC [i.e., $(\text{C}_4\text{H}_{12}\text{N}_2)\text{Cu}_2\text{Cl}_6$], see Glazkov *et al.*, Phys. Rev. B 85, 054415 (2012).

For KCuCl_3 , the situation is more complicated indeed, as there is no direct evidence

of significant anisotropy effects (to the best of our knowledge, no high-resolution ESR experiments were conducted for different magnetic field directions). However, we would like to point out that (a) the symmetry is equivalent to TlCuCl_3 , allowing for biaxial anisotropy, and (b) the pressure-induced staggered magnetization is oriented at a particular angle. This is an indirect evidence for complex anisotropy. We would like to stress that DTN is fundamentally different; the in-plane anisotropy is absent in DTN based on the lattice symmetry. We have expanded the symmetry discussion and added more references to avoid ambiguity.

c. Figure 2c shows the change of lattice parameters with pressure. The results obtained in a gas and clamp pressure cell do not seem to align with each other. Considering that the pressure in the gas pressure cell was determined in situ via the manometer, it is very likely that the actual pressure deviates from the measured value. To prevent any confusion, I would suggest disregarding the results obtained in the gas pressure cell.

The clamp-cell results are all above the critical pressure indeed, but actually there is no controversy here. There is a small overlap between the gas-cell and clamp-cell results around the structural transition, and the corresponding lattice constant discontinuity can be seen in the gas-cell data alone.

However, we have realized that the corresponding figure was too much loaded with various symbols, thus provoking some confusion. In the revised manuscript we cleared up the figure, also using the same symbol for both cell types to improve readability.

d. To support their conclusion (i.e., the onset of the antiferromagnetic order induced by the pressure) more robustly, can the authors consider further supporting measurements, such as neutron diffraction, or muon spin relaxation, or heat capacity with He-3 or dilution refrigerator?

We would like to thank the Referee for this very valuable suggestion. As mentioned in the Resubmission Letter to the Editor, this very point made by both Referees has motivated us to perform additional measurements. Following the Referee's recommendation we have attempted high-pressure specific-heat measurements with the help of M. Nicklas at Max Planck Institute for Chemical Physics of Solids, Dresden, Germany, but unfortunately this technically challenging experiment did not work out. Our previous μSR measurements of DTN at the Paul Scherrer Institute (Villigen, Switzerland) also did not lead to any conclusive statements, revealing weak interaction between the sample and muons.

On the other hand, high-pressure ultrasound measurements with the help of A. Hauspurg and S. Zherlitsyn (Dresden High-Magnetic Field Laboratory, Helmholtz-Zentrum Dresden-Rossendorf, Dresden, Germany) appeared extremely helpful. This method is known as a very powerful tool to probe magnetic phase transitions in strongly correlated electron systems (see Ref. 35 in the main text). By using a high-pressure ultrasound technique we were able not only to confirm the data obtained by high-field TDO measurements, but also to reveal the onset of long-range antiferromagnetic order in DTN in zero field. This is exactly the behavior one would expect while driving DTN as an easy-plane antiferromagnet towards the semiclassical regime. We further would like to stress that the obtained data are in a perfect agreement with results of zero-pressure ultrasound studies (Ref. 33), where the onset of the field-induced ordering in DTN was studied in detail.

Sincerely,
Kirill Povarov
(on behalf of the authors)

Reply to Referee 2

The authors investigated the pressure-driven gapped-to-gapless phase transition in $\text{NiCl}_2 \cdot 4\text{SC}(\text{NH}_2)_2$ (DTN). They suggested an occurrence of antiferromagnetic (AFM) order induced by pressure, and quantitatively described the pressure-driven evolution of the critical fields and spin Hamiltonian parameters in DTN. They investigated the DNT under pressure by means such as tunnel diode oscillator (TDO) susceptibility measurements, electronic spin resonance (ESR) spectra, neutron diffraction studies. However, DTN, which is an old system, has been intensively investigated previously. Thus, it is lack of broad interests.

We would like to thank Referee for the careful reading and deep assessment of our manuscript. However, we certainly disagree on the last point. The quantum magnet DTN is still attracting intense interest for the reason of being a very clean and easy-to-interpret model system with exceptionally high spin symmetry. In previous studies, magnetic field was applied as a tuning parameter, driving DTN through the quantum critical regime described by $z = 2$ universality class. Contrary to that, we show that DTN is a perfect system for the realization of pressure-induced $z = 1$ -type quantum criticality keeping the simple undistorted structure. Thus, it is not just about DTN as material, but rather about a new type of quantum critical phenomenon in a high-symmetry spin system, where pressure is used as a tuning parameter. What is also important is that DTN is a promising platform, equally appealing to experimentalists (the material is easy to obtain and handle, the quantum critical regime is readily accessible) and theorists (the Hamiltonian is simple, almost minimum for this kind of problem). We have modified the text in order to stress these points more clearly.

Besides, there are some other problems with this manuscript.

1. The authors declared that the quantum-disordered state (QD) is suppressed by the pressure, and an AFM ordering is induced. However, the experiment just shows the suppression of QD, but not the emergence of AFM. As we know, at the quantum critical point, the suppression of a phase sometimes is sometimes accompanied by the appearance of another new phase. We admit the suppression of the QD. However, the phase above P_c need to be determined very carefully. Therefore, how to undoubtedly prove it is an AFM phase? And what is the magnetic state above 6 kbar?

We would like to thank the Referee for this very important comment. As mentioned, this has motivated us to perform additional measurements. Most importantly, using ultrasound as tool to probe spin ordering in DTN, we not only confirmed the data obtained by our TDO measurements, but (more importantly) identified the nature of the high-pressure magnetic transition (see our response to the Referee 1). As for the magnetic state above 6 kbar, not much known so far, apart from the fact that it has a similar saturation field. Most likely it is some version of simple Néel order with extra modulations imposed by the periodically deformed lattice. We would like to reiterate, that from the perspective of the tentative $z = 1$ quantum critical point (QCP) that we have found at P_c , the presence of the structurally distorted phase above the higher P_{irr} is a small nuisance. Nonetheless, what happens at P_{irr} has no direct relevance to the magnetism near P_c . Detailed investigation of structural and magnetic properties of this high-pressure phase we leave for future studies. We have introduced some changes in the text to sharpen focus on the magnetic transition at P_c rather than on the structural transition at P_{irr} .

2. In the Figure 6 of the ESR spectra, there is no obvious change before and after the critical pressure P_c . The ESR is performed at $T = 4$ K. It seems that this measurement has no relation with the QCP discussed here.

Indeed, the ESR response of DTN varies smoothly across the phase transition. This has to do with the fact that ESR probes the dynamic structure factor at a wavevector far away from the critical one. The existing smooth dependence is governed by the continuous evolution of the spin-Hamiltonian parameters rather than by the change of the ground state. Unfortunately, the current theoretical framework is limited to a semi-quantitative comparison only. To the best of our knowledge, currently there are no tools to simultaneously treat the thermal and quantum fluctuations at high energies (the $q = 0$ wavevector corresponds to the top of the band in Fig. 1) in the nearly critical regime. On the other hand, taking into account the Referee's remarks, we have substantially reworked the ESR section to make it lighter and easier to follow for the reader. All comparisons to the theory have been moved to the Supplemental Material.

3. In Figure 3 of TDO, the H_{c1} disappears between 3.2 kbar and 4.2 kbar, while the H_{c2} persists through the pressure-induced structural phase transition. Only at 6.0 kbar, the H_{c2} splits into two coexisted peaks. what does the coexistence mean? Since TDO is corresponding to the magnetization, does it mean that the coexistence of two kinds of magnetic ordering? Furthermore, what are exactly these two magnetic ordering phases?

The “double” transition at 6 kbar TDO data is an apparent manifestation of the first-order character of the structural distortion. It means that two types of structural domains coexist within the same crystal, with different magnetic properties and, thus, different saturation fields. The very same sample gives a single H_{c2} peak at other pressures, which are below or above P_{irr} .

We can rather straightforwardly assign these peaks as belonging to the upper critical field for the simple Néel order (peak at higher field) or the upper critical field for a yet-unknown magnetic state in the structurally distorted regime (peak at lower field). They fall on the corresponding linear $H_{c2}(P)$ dependences (Fig. 6 in the main text).

4. Due to the deficiency of spectra between 3.2 kbar and 4.2 kbar in Figure 3, it is hard to determine the P_c from experiment within 1 kbar.

This point has some overlap with our reply to Comment 6. The existence of a self-consistent description of H_{c1} as function of D , J_a , and J_c (the RPA ansatz) allows us to make use of the whole array of data points, not only the ones in the direct vicinity of P_c . This allows us to obtain a somewhat “sharper” estimation of the tentative P_c range, so that the resulting error bar of 0.3 kbar is of statistical nature.

We would also like to note that the inclusion of the ultrasonic data leads to a more detailed set of data, reassuring the initial result being correct.

5. This DTN exhibits an irreversible distortion of the lattice at higher pressures exceeds 6 kbar. Is it reversible or irreversible below 6 kbar? Since the irreversible distortion, these experimental should be performed on different samples. Thus, how to guarantee the same quality of these samples used?

It seems we have not discussed these important issues clear enough in the original manuscript. The behavior of DTN below 6 kbar is fully reversible. All the experi-

mental data presented was collected with increasing the pressure stepwise. We used different “virgin” samples in different experiments. Overall, the sharpness of the magnetic transition at all pressures reassures that the crystals do not suffer mechanical damage. We have made corresponding modifications in the text to clarify these points.

6. In Figure S4, RPA ansatz fit gives that $P_c = 4.3$ kbar. However, in figure 3, at 4.2 kbar, the H_{c1} has been disappeared. What fact caused this large deviation of P_c ?

There is no contradiction here. The RPA ansatz is essentially a fit having the value of P_c as an open parameter. As it optimizes not only the point at which H_{c1} disappears, but the non-zero values of H_{c1} as well, the result of the fit might be a bit away from the “apparent” gap closing. Nonetheless, the compliance with the first $H_{c1} = 0$ point naturally has a significant weight in the fit procedure, and with the statistical error bar of 0.3 kbar the fit result is actually in excellent agreement with the gap closure at 4.2 kbar.

Summary, the author investigated the DTN under pressure by using several challenging measurement techniques, such as TDO, ESR spectra, neutron diffraction under pressures. These devices are very excellent in the area of experimental study. However, it seems that only one experimental measurement is related to the QCP discussed. Moreover, this measurement cannot give the solid proof for the new phase above the P_c . Due to these reasons, I cannot recommend the publication of this article in this journal.

We would like to reiterate that different measurement techniques touch upon some different aspects of the quantum critical regime we are discussing. The TDO measurements (and now as well the ultrasound experiments) allow us to identify the magnetic QCP in the first place. Neutrons are used to confirm that the QCP of interest is not related to any structural distortion (that occurs at the significantly higher pressure P_{irr}). Finally, the ESR data is used to qualitatively crosscheck the validity of the pressure-induced Hamiltonian parameters evolution from the spin-dynamics perspective. We would also like to note that the presence of the long-range ordered phase between P_c and P_{irr} at zero field is now directly confirmed by the ultrasound data.

We hope that we have resolved all the ambiguities related to the data and the discussion in the updated version of the paper. We also hope that with these changes and additional data presented in favor of our conclusions, the Referee would find our manuscript suitable for publication in Nature Communications.

Sincerely,
Kirill Povarov
(on behalf of the authors)

REVIEWERS' COMMENTS

Reviewer #1 (Remarks to the Author):

I would like to thank the authors for considering my comments and making the changes. My main concern has been addressed by the fact that the additional high-pressure ultrasound measures confirm the onset of the pressure-induced antiferromagnetic order in DTN. I believe that this comprehensive work establishes DTN as an ideal experimental platform for investigating quantum phase transition with $z=1$ universality class and thus recommend the publication of the manuscript as is.

Reviewer #2 (Remarks to the Author):

I have read the revised manuscript and the reply letter from the authors. They have made substantially revision to this article. In particular, they added high-pressure ultrasound measurements, which can further support their discussion and conclusion. Moreover, the concerning discussion are also revised carefully. Therefore, I would like to support publishing the manuscript in this journal.